# Inhibiting Effects of Ginger and Rosemary on the Formation of Heterocyclic Amines, Polycyclic Aromatic Hydrocarbons, and Trans Fatty Acids in Fried Pork Balls

**DOI:** 10.3390/foods11233767

**Published:** 2022-11-23

**Authors:** Xiaomei He, Baichenyang Li, Xiaoyan Yu, Yuan Zhuang, Changmo Li, Lu Dong, Yan Zhang, Shuo Wang

**Affiliations:** 1Key Laboratory of Food Nutrition and Safety, Ministry of Education of China, Tianjin University of Science and Technology, Tianjin 300457, China; 2Tianjin Key Laboratory of Food Science and Health, School of Medicine, Nankai University, Tianjin 300071, China

**Keywords:** heterocyclic amines, polycyclic aromatic hydrocarbons, trans fatty acids, inhibitory profile, antioxidant

## Abstract

The preparation of fried meat products is prone to the formation of large amounts of heterocyclic amines (HCAs), polycyclic aromatic hydrocarbons (PAHs), and trans fatty acids (TFAs), which are potential risks to human health. Spices contain natural antioxidants that can inhibit the oxidation of fats and oils and the formation of hazardous substances. In this experiment, the effect of adding different levels (0.25%, 0.75%, 1.25%) of ginger or rosemary during meatball preparation on the formation of HCAs, PAHs and TFAs in fried pork balls was investigated. The results showed that the addition of ginger and rosemary reduced the content of HCAs in fried pork balls compared to the control group (no added spices). The inhibition of total HCAs was 63% when 0.25% ginger was added, while the total HCA content was reduced to 59% when 0.25% rosemary was added. The addition of 0.25% and 0.75% rosemary reduced the PAH content in fried pork balls by 30% and 35%. In addition, ginger and rosemary showed significant inhibition of C20:1 11t in TFAs, with a maximum inhibition rate of 40%. Therefore, adding appropriate levels of ginger or rosemary to fried pork balls could simultaneously inhibit the formation of HCAs, PAHs, and TFAs.

## 1. Introduction

Deep-fried meat products have an attractive color and unique flavor, so they are often favored in daily life. However, numerous studies have shown that deep-fried meat products contain heterocyclic amines (HCAs), polycyclic aromatic hydrocarbons (PAHs), trans fatty acids (TFAs), and other harmful substances [1,2]. IQ in HCAs is classified as a probable human carcinogen (Class 2A) by the International Agency for Research on Cancer (IARC), and MelQ, MelQx, PhIP, AαC, MeAαC, Trp-P-1, Trp-P-2, and Glu-P-1 are also labeled as possible human carcinogens (Class 2B) [3]. Hundreds of PAHs are found in the environment and have been classified by the International Agency for Research on Cancer. Benzo[a]pyrene has been categorized as group 1 (carcinogenic to humans). Dibenzo[a,h]anthracene (DahA) and dibenzo[a,l]pyrene have been classified as group 2A (possibly carcinogenic to humans). Benzo[a]anthracene (BaA), benzo[b]fluoranthene (BbF), and benzo[j]fluoranthene (BjF) are listed as group 2B (possibly carcinogenic to humans) [4]. The PAHs present in food are benzo(a)anthracene (BaA), chrysene (Chr), benzo(b)fluoranthene (BbF), and benzo(a)pyrene (BaP). The formation of PAHs in food is mainly due to pyrolysis of the organic matter thermal-synthesis reaction and incomplete combustion of the heat source [5]. In addition, for fried foods, the quality and type of oil used for frying has an important influence on the formation of PAHs in fried foods, where blended oils are more stable than single oils [6]. TFAs in people’s daily diet are mainly formed during the processing of food, such as the hydrogenation and heat treatment of fats and oils, and research has also shown a link between TFAs and cancer and type 2 diabetes [7]. Since these hazards are not only carcinogenic but also cause other diseases, there is a great deal of interest in measures to effectively reduce the production of harmful substances in fried meat products.

Scholars have proposed a number of measures to retain the characteristics of fried meat products while effectively reducing the risks, including through improved processing methods, low temperature and short cooking times, the use of appropriate frying oils, and the addition of antioxidants [8,9]. As the safety of industrial synthetic antioxidants is still in doubt, the inhibition of natural antioxidants against hazards generated during food processing has become one of the focal points of research. The main sources of natural antioxidants are herbs and spices, in addition to some fruits that also contain antioxidant substances. Some have been shown to inhibit the production of hazardous substances during the processing of meat products. Wang et al. studied the effect of eight pure phenolic compounds, including green tea extract, on the formation of PAHs in charcoal grilled chicken wings and found that these phenolic compounds inhibited PAH formation from 15.1% to 54.5% [10]. Meurillon et al. reported that resveratrol completely inhibited MeIQ in beef patties and reduced the formation of MeIQx and PhIP by 40% and 70%, respectively [11]. However, there are few reports on the synchronous inhibition of multiple hazards in meat products by natural active substances.

Ginger and rosemary are two spices with unique flavors that are often used in daily food preparation. The active ingredient in ginger is mainly curcumin, which has strong antioxidant activity, while rosemary contains substances with antioxidant capacity such as syringic acid, rosmarinic acid, syringol, etc. [12,13]. In addition, studies have shown that rosemary extract has high heat resistance, with a loss of only 6% at a frying temperature of 190 °C, and could effectively inhibit the production of polar substances during frying [14]. Because of the above properties of ginger and rosemary, these two spices were chosen as the subjects of this experiment. Currently, the simultaneous inhibitory effects of ginger and rosemary on HCAs, PAHs, and TFAs in meat products have not been fully investigated. The presence and hazard of HCAs, PAHs, and TFAs have become a major concern for both consumers and researchers. Hence, the effect of adding different levels (0.25%, 0.75%, 1.25%) of ginger or rosemary on the content of HCAs, PAHs, and TFAs in fried pork balls was investigated in this experiment. The purpose was to provide an effective measure for the reduction of HCAs, PHAs, and HCAs in fried pork balls.

## 2. Materials and Methods

### 2.1. Materials and Reagents

Lean pork was purchased from a local market (Tianjin, China). Visible fat and tendons in the pork were trimmed off and stored at 4 °C. Ginger powder and rosemary powder were commercial spices produced by McCormick Company (Hunt Valley, MD, USA). The oil used for frying was commercial soybean oil produced by Shandong Luhua Group Co. (Yantai, China).

The standards MeIQ (2-Amino-3,4-dimethylimidazo [4,5-f]quinoline), IQx (2-amino-3-methylimidazo [4,5-f]-quinoxaline), MeIQx, 4,8-DiMeIQx, 7,8-DiMeIQx (2-amino-3,7,8-[4,5-f]quinoxaline), PhIP, Harman, Norharman, Trp-P-2 (3-Amino-1-methyl-5H-pyrido [4,3-b]indole), AαC (2-Amino-9H-pyrido [2,3-b]indol), MeAαC (2-Amino-3-methyl-9H-pyrido [2,3-b]ind-ol), IQ, BaA (Benz[a]anthracene), Chr (chrysene), BaP (benzo[a]pyrene), BbF (benzo[b]fluorant-hene), C16:1 9t, C18:1 9t, C18:1 11t, C18:2 9t 12t, and C20:1 11t were purchased from Sigma-Aldrich (St. Louis, MO, USA). Thiobarbituric acid (TBA), tetraethoxypropane (TEP), anhydrous sodium bisulfate, potassium ferrocyanide, formic acid, hydrochloric acid, n-hexane, Isooctane, cyclohexane, dichloromethane, ethyl acetate, acetonitrile, and ethanol were purchased from Aladdin Biochemical Technology Co., Ltd. (Shanghai, China). Gallic acid and 1,1-diphenyl-2-trinitrophenylhydrazine (DPPH) were purchased from Macklin Biochemical Co., Ltd. (Shanghai, China). Petroleum ether, potassium hydroxide, ammonia water, sodium carbonate, magnesium sulfate, and sodium acetate were purchased from Tianjin Baishi Chemical Industry Co., Ltd. (Tianjin, China). Folin–Ciocalteu reagent and ethylenediaminetetraacetic acid (EDTA) were purchased from Solarbio Life Science Co., Ltd. (Beijing, China). Zinc acetate and sodium hydroxide were purchased from Sinopharm Chemical Reagent Co., Ltd. (Shanghai, China). Methanol was purchased from Fisher Scientific (Loughborough, UK). Solid phase extraction C-18 column (500 mg, 6 mL) was purchased from Bonna-Agela Technologies Co., Ltd. (Tianjin, China). Solid phase extraction UPAH columns (500 mg, 6 mL) were purchased from Zhongpu Science (Fuzhou, China). Solid phase extraction Oasis^®^ MCX column (60 mg, 3 mL) was purchased from Waters Corporation Co. (Milford, MA, USA).

### 2.2. Preparation and Cooking Process of Pork Balls

Tenderloin in the fresh pork was cleaned, and the excess fat and tendons were removed and ground by a meat grinder. Then 0.25%, 0.75%, or 1.25% of ginger or rosemary were added to the ground meat, mixed thoroughly, and placed in a refrigerator at 4 °C for 6 h. A control group (no spices added to the minced meat) was also set up. A total of 20.0 g (accurate to 0.01 g) of mince was then weighted and shaped into balls using molds. The fryer was preheated, and the meat was fried using soybean oil at 180 °C for 3 min. Excess oil was absorbed using absorbent paper to record the final weight and calculate the cooking loss rate. Cooking loss rate calculations were as follows: [(Wr −Wc) ÷ (Wr ×100)], where Wr is the weight of raw pork balls and Wc is the weight of cooked pork balls after frying. The fried pork balls were stored at −18 °C for further analysis and testing.

### 2.3. Moisture, pH, and Fat Content Analysis

The moisture content of fried pork balls was determined according to the Chinese national standard method (GB5009.3, 2016). The pH of fried pork balls was determined according to the Chinese national standard method (GB5009.237, 2016). The fat content was determined by the Soxhlet extraction method.

### 2.4. Determination of Thiobarbituric Acid Reactive Substances (TBARS) Values

The TBARS values in the samples were determined by referring to the method of Jin et al., with slight modifications [15]. A standard curve was plotted using 20 μg/L, 50 μg/L, 100 μg/L, 200 μg/L, and 500 μg/L 1,1,3,3-tetraethoxypropane (TEP) solutions. To 2 g of meat sample, 10 mL of 7.5 g/100 mL TCA (containing 0.1% EDTA) was added, and the sample was homogenized with a 5000 rpm high-speed homogenizer for 30 s; then, to 3 mL of filtrate, 3 mL of 0.02 mol/L TBA solution as added, mixed well and kept in a boiling water bath at 100 °C for 1 h. After removing the samples and cooling them to room temperature, the absorbance values were measured at 532 nm using an enzyme marker (Varioskan LUS, Thermo, Waltham, MA, USA). The TBA values of the samples were calculated by comparison with the TEP standard curve, and the results were expressed as mg MDA/kg (in meat samples) (MDA is malondialdehyde).

### 2.5. Determination of Total Phenolic Content (TPC) and Antioxidant Capacity of Spices

Total phenolic content was determined using Folin–Ciocalteu agent. Then, 1 g of spices was added to 30 mL of ethanol and water (6:4, *v/v*) and extracted ultrasonically at 70 °C for 1 h; the sample was then centrifuged at 6000× *g* and the supernatant was extracted. The supernatant was diluted appropriately, and 0.5 mL of the diluted extract was mixed with 1 mL of Folin–Ciocalteu reagent for 5 min; then, 1 mL of 10% Na_2_CO_3_ was added and allowed to react under light-proof conditions for 1 h. The absorbance was measured at 760 nm wavelength using an enzyme marker (Varioskan LUS, Thermo, Waltham, MA, USA). The standard curve was prepared using 40 μg/mL, 60 μg/mL, 100 μg/mL, 150 μg/mL, and 200 μg/mL of gallic acid solution. The results were expressed as gallic acid equivalents (GAE) per gram dry weight (dry weight). The supernatant obtained from spice extraction was freeze-dried to obtain spice extracts and configured into different concentrations of sample solutions to determine the antioxidant capacity of spices. The scavenging capacity of spices for DPPH radicals referred to the method of Sethi, with slight modifications [16]. The scavenging capacity of ABTS radicals referred to the method of Villano, with slight modifications [17].

### 2.6. Determination of HCAs

The detection method of Wang et al. was used [18]. A total of 1 g of minced meat sample was weighed in a 50 mL centrifuge tube, and 10 mL of 0.1 mol/L NaOH was added and vortexed for 5 min; then, 10 mL of acetonitrile was added and vortexed for 1 min, and 6 g of MgSO_4_ and 1.5 g of sodium acetate were added and vortexed for 5 min. Finally, the mixture was centrifuged at 2300× *g* for 5 min, and the supernatant was collected and the and solid phase extraction performed. The MCX column was activated with 3 mL of methanol and 3 mL of 0.01 mol/L hydrochloric acid solution. The entire collected supernatant was transferred to the activated column so that the target was adsorbed in the packing of the solid extraction column. After the supernatant transfer was completed, 3 mL of 2% formic acid water, 9 mL of deionized water, and 3 mL of methanol were used to wash the debris. The MCX column was then lavaged with 3 mL of methanol/ammonium hydroxide (9:1, *v*/*v*), and the eluate was collected. The eluate was nitrogen blown dry and then re-dissolved with 1 mL of methanol. The resolution was filtered through an organic microporous membrane (0.22 μm) and placed in a sample vial to be analyzed by UPLC-MS/MS (ACQUITY/XevoTQ-S micro, Waters, Milford, MA, USA).

The separation of HCAs was performed on an ACQUITY UPLC BEH C18 column. The temperature of the column was set at 45 °C. The mobile phases, A and B, were ultrapure water and acetonitrile, respectively. The flow rate was set at 0.5 mL/min, and the injection volume was 1 μL. The mobile phase gradient program was: 0–0.5 min, 90% A; 0.5–10 min, 50–90% A; 10–10.1 min, 50–90% A; 10–15 min, 90% A. ESI was performed in positive-ion mode; MRM mode was used for MS/MS detection. The optimal ionization source working parameters were as follows: nebulizer pressure, 40 psi; capillary voltage, 900 V. The drying gas flow rate and temperature were 800 L/Hr and 400 °C, the cone gas flow rate was 60 L/Hr, and the ion source temperature was 150 °C. The validation results of the method are shown in Appendix A.

### 2.7. Determination of PAH4

The detection method of Lee et al. was used, with slight modifications [19]. A total of 5 g of minced meat sample was weighed in a 50 mL centrifuge tube, and 20 g of anhydrous sodium sulfate and 20 mL of cyclohexane-ethyl acetate (1:1, *v*/*v*) were added. After vortex shaking for 1 min and ultrasonic extraction for 15 min, the sample was then centrifuged at 2300× *g* for 10 min, and the supernatant was aspirated and blown under nitrogen until nearly dry. Then, 5 mL of 1.5 mol/L potassium hydroxide-ethanol solution was added to the ultrasonic solution for 5 min, followed by a water bath at 70 °C for 2 min, and the centrifuge tube was removed and cooled rapidly. Then, 4 mL of ultrapure water and 5 mL of hexane was added, vortexed, and shaken for 2 min; it was then centrifuged at 9400× *g* for 2 min, and the upper layer of the hexane extract was aspirated for purification. The purification step was to add 1 g of anhydrous sodium sulfate to the UPAH column and then activate the UPAH column with 5 mL of dichloromethane and 5 mL of n-hexane in turn (control the liquid drop rate of 1–2 drops/s); all supernatants were aspirated over the activated UPAH column, washed with 4 mL of n-hexane to remove impurities, and finally eluted with 5 mL of dichloromethane-ethyl acetate mixture. The eluate was collected and concentrated to near dryness by nitrogen blowing at 4 °C. A total of 0.5 mL of the acetone-isooctane mixture was added, vortexed, mixed to fully dissolve the PAHs, and analyzed by GC-MS (7890B-5977A, Agilent, Santa Clara, CA, USA).

The separation of PAHs was achieved using an HP-5MS (30 m × 0.25 mm × 0.25 μm) column with programmed temperature rise. The carrier gas was helium at a flow rate of 1.0 mL/min. A total of 1 μL of the extract was automatically injected using a non-split mode with an inlet temperature of 320 °C. The column temperature was programmed with a programmed ramp, and the starting temperature of the column was set at 0.5 °C. The column temperature was programmed to start at 80 °C, held for 1 min, and ramped up to 220 °C at a rate of 4 °C/min, and then to 280 °C at a rate of 20 °C/min and held for 10 min. The transfer line and ion source temperatures were set to 280 °C and 250 °C, respectively, using an electron bombardment (EI) ionization source with an electron energy of 70 eV and a solvent delay time of 3 min. The analysis of PAHs was performed in selective ion monitoring mode (SIM mode). The validation results of the method are shown in Appendix A.

### 2.8. Determination of TFAs

The TFAs in the samples were in accordance with the method of Zribi et al., with slightly modifications [20]. The fat was extracted by Soxhlet extraction, and 60 mg was taken to prepare the fatty acid methyl esters. First, 4 mL of hexane was added for dissolving, followed by 0.2 mL of 2 mol/L potassium hydroxide-methanol solution; the sample was vortexed, mixed for 1 min, and left until the mixture in the tube was clarified. The supernatant was filtered through an organic microporous membrane (0.22 μm) and placed in a sample bottle for GC analysis (GC-2010, Shimadzu, Kyoto, Japan).

The GC analysis was performed on a gas chromatograph equipped with a hydrogen flame ionization detector and a split injector. The samples were separated using an HP-88 capillary column (100 m × 0.25 μm × 0.2 μm). The inlet temperature was 250 °C, and the detector temperature was 250 °C. The sample was manually injected at a split ratio of 1:29 in a volume of 1 μL. The carrier gas was helium at a flow rate of 1.0 mL/min. The determination was performed by gas chromatography. The column temperature was programmed to start at 120 °C and held for 4 min, then increased to 175 °C at a rate of 10 °C/min and held for 6 min, and then increased to 210 °C at a rate of 5 °C/min and held for 12.5 min.

### 2.9. Inhibitory Rates of HCAs, PAH4, and TFAs

Inhibition rate (%) calculations were as follows: [(Ac−At) ÷ (Ac ×100 )], where Ac is the concentration of hazards in the control sample and At is the concentration of hazards in the pork balls containing spices.

### 2.10. Statistical Analysis

The results were expressed as mean ± standard deviation (SD). SPSS (v. 26.0) was used for the analysis of experimental data. Origin Pro (v. 2021) was used for drawing figures.

## 3. Results and Discussion

### 3.1. Antioxidant Capacity and TPC of Spices

The antioxidant capacities of ginger and rosemary as well as the total phenolic content are shown in Table 1. The antioxidant activities of DPPH radicals and ABTS radicals were expressed as IC_50_ values (mg/mL). In the DPPH radical scavenging capacity assay, the IC_50_ values were 0.26 mg/mL for ginger extract and 0.03 mg/mL for rosemary extract. The IC_50_ values of the ABTS radical scavenging capacity assay of ginger and rosemary extract were 4.60 mg/mL and 0.93 mg/mL, respectively. The results showed that rosemary had a stronger scavenging capacity for DPPH radicals and ABTS radicals than ginger. In addition, the total antioxidant capacity of the two spice extracts measured by the FRAP method was 0.58 ± 0.01 μmol/mL and 3.74 ± 0.04 μmol/mL, respectively, indicating once again that the antioxidant capacity of rosemary was stronger than that of ginger. From the data in Table 1, it can be observed that the total phenolic content of the two spices differed significantly, with the total phenolic content of ginger being 9.90 ± 0.27 mg GAE/g and that of rosemary being 61.34 ± 1.23 mg GAE/g. The results indicate that the higher the total phenolic content, the stronger the antioxidant activity. Many studies have also reported a correlation between total phenolic content and antioxidant activity. Erkan et al. investigated the phenolic content and antioxidant activity of blackseed essential oil and rosemary extract, and the results showed that the phenolic content of rosemary was higher than blackseed essential oil and the antioxidant activity of rosemary extract was also higher [21]. Noreen et al. showed experimentally that phenolic compounds were the main contributors to the antioxidant activity [22].

### 3.2. Moisture, pH, Fat Content, and Cooking Loss of Deep-Fired Pork Balls

Table 2 shows the effect of ginger and rosemary on the cooking loss rate, moisture content, pH, and fat content of deep-fried pork balls at different addition levels. The addition of spices had a significant effect on the cooking loss values of the deep-fried pork balls compared to the control group, and the cooking loss values in the samples decreased as the amount added increased. This result is in agreement with the findings of Oz, who showed that the use of black cumin in meatball preparation significantly reduced the cooking loss values [23]. It has been reported that some substances such as myogenic fibers, collagen, and lipids are lost during cooking, in addition to water [24]. The moisture content of deep-fried pork balls treated with spices increased significantly (*p* < 0.05) compared to the control group, and this result could be attributed to the absorption of some of the free water in the pork balls by the spices as a dry substance. It is noteworthy that a negative correlation between cooking loss and moisture content was shown, which is logical. From the values shown in Table 2, it can be observed that the pH values of all samples were slightly acidic, with values ranging from 6.47 to 6.72. The addition level of spices had no significant effect on the pH of deep-fried pork balls (*p* > 0.05). Kilic et al. also found that the added level of turmeric did not affect the pH of deep-fried chicken pork balls [25]. The fat content of deep-fried pork balls in this experiment ranged from 6.27 to 7.57 g/100 g (as shown in Table 2). The level of spices added had no significant effect on the fat content.

### 3.3. Effect of Spice Addition on the TBARS of Deep-Fried Pork Balls

The production and storage of meat products are often accompanied by lipid oxidation, and numerous studies have shown that the lipid oxidation of meat is one of the causes of the quality of meat products. In this experiment, TBARS values were used to characterize the content of lipid oxidation secondary products [26]. The TBARS values of the samples are shown in Figure 1. The TBARS value for the control group was 0.44 ± 0.01 mg MDA/kg. The addition of ginger and rosemary significantly reduced the TBARS values of deep-fried pork balls (*p* < 0.05), with TBARS values ranging from 0.11 to 0.15 mg MDA/kg, while the type of spice also had a significant effect on the TBARS values (*p* < 0.05). The addition of rosemary to pork balls was more effective in reducing lipid oxidation than the addition of ginger, probably due to the stronger antioxidant capacity of rosemary than ginger. Numerous studies have also shown that the addition of antioxidant substances to meat products is effective in reducing their TBARS values. The addition of hawthorn extract to chicken and beef pork balls has been reported to be effective in reducing lipid oxidation [27]. The addition of black cumin to deep-fried pork balls also significantly reduced TBARS values, and the TBARS values in the samples decreased with increasing addition [23].

### 3.4. Effect of Spices on the Formation of HCAs in Deep-Fried Pork Balls

Table 3 shows the HCA content in the control and spice-added fried pork balls. In this study, the major contributors to the total HCA content were IQ, Harman, Norharman, and IQx, while other HCAs, including MeIQ, MeIQx, MeAαC, 4,8-DiMeIQx, 7,8-DiMeIQx, PhIP, Trp-P-2, and AαC, were found in low levels in fried pork balls. The amount of IQ in deep-fried pork balls under ginger and rosemary treatment was significantly lower than the control (4.16 ± 0.16 ng/g), and the least amount of IQ was obtained when 0.25% ginger was added (0.43 ± 0.12 ng/g), which was inhibited by 90%. The rest of the experimental groups also showed inhibition rates between 58% and 84%, indicating that the addition of ginger and rosemary to deep-fried pork balls effectively inhibited the production of IQ. This result is consistent with the results of other studies. Lu et al. found that ginger was effective in inhibiting IQ in deep-fried chicken pork balls [1]. Murkovic et al. showed that rosemary can reduce IQ in meat products [28]. Harman was similar to the IQ results, the Harman content in the spice-treated samples decreased significantly (*p* < 0.05) compared to the control group (2.02 ± 0.18 ng/g), with inhibition rates ranging from 43% to 81%. Among them, the addition of rosemary had a significant effect (*p* < 0.05) on the content of Harman, and the best inhibition of Harman was achieved when 0.25% rosemary was added. Unlike IQ and Harman, the addition of ginger and rosemary to pork balls had no inhibitory effect on the production of Norharman and showed a significant increase (*p* < 0.05) compared to the control (1.44 ± 0.21 ng/g). Both the level and type of spices added had a significant effect (*p* < 0.05) on the content of Norharman in fried pork balls, with rosemary-treated pork balls having a higher amount of Norharman than that of ginger. This result is similar to other reports, where Zeng et al. showed a significant increase in Norharman content after the addition of Sichuan pepper to beef patties before cooking, and also suggested that the reason for this result could be the presence of Norharman or its precursors in Sichuan pepper [29]. In addition, it has been demonstrated that phenolic acids in spices during the processing of meat products have an enhancing effect on the production of Norharman [30]. The enhancement of Norharman by ginger and rosemary may be due to the phenolic richness, and the stronger enhancement of Norharman by rosemary may be due to the higher total phenolic content of rosemary than ginger. The addition of ginger and rosemary was also effective in inhibiting the formation of IQx in fried pork balls.

The total HCA content in the control group of this study was 8.84 ± 0.43 ng/g, a result that is generally consistent with the data reported in the literature. Shin et al. found 13.91 ± 1.81 ng/g of HCAs in fried pork [31]. Oz et al. also reported that the total HCA content of control beef patties fried at 175–225 °C for 10 min per side ranged from 1.40 to 37.81 ng/g [32]. The results of this study showed that the addition of spices reduced the total HCA content in fried pork balls, with inhibition rates ranging from 5% to 63%. Numerous studies have also shown that the addition of spices such as black cumin [23], galangal [33], and chili pepper [34] to meat products could reduce the content of total HCAs. In this study, the addition of 0.25%, 0.75%, and 1.25% ginger inhibited 63%, 48%, and 11% of total HCAs in pork balls, respectively; the addition of rosemary to pork balls at the same dose as ginger resulted in 59%, 32%, and 5% inhibition of total HCAs, respectively. This result indicates that the inhibition of HCAs in fried pork balls by spices was dose-dependent, with lower levels of spice addition showing higher inhibition effects. This result is similar to that reported by Zeng et al., who showed inhibition of 46%, 35%, and 24% of total HCAs when 0.5%, 1.0%, and 1.5% of chili peppers were added to roast beef patties, respectively, and suggested that it may be that some components of chili peppers have a facilitative effect on HCA production [35]. In addition, Damasius et al. also indicated that some phenolic compounds and spice extracts promote the production of HCAs [36]. Therefore, it may be that some phenolic compounds in ginger and rosemary promoted the formation of HCAs in this study. Furthermore, antioxidants not only have antioxidant effects but also promote oxidation, and their effects depend on the concentration and conditions of the antioxidants [37].

### 3.5. Effect of Spices on the Formation of PAH4 in Deep-Fried Pork Balls

Table 4 shows the content of PAH4 in fried pork balls. BaA, Chr, and BaP were quantified in the study, and BbF was not detected in any sample. The PAH4 content in the control group in this study was 9.34 ± 0.27 ng/g. The results showed that ginger had no significant effect on the content of Chr, BaP, and total PAH4 in deep-fried pork balls. The addition of different concentrations of rosemary did not have a reducing effect on the BaA content of the fried pork balls compared to the control group. When 0.25% and 0.75% rosemary were added, Chr, BaP, and PAH4 contents in fried meatballs were significantly reduced compared to the control group (*p* < 0.05), where the addition of 0.75% rosemary had the best inhibition effect on Chr, BaP, and PAH4, with 41%, 42%, and 35% inhibition rates, respectively. The analysis revealed that the type of spice had a significant effect on PAH4 (*p* < 0.05), and rosemary had a better inhibitory effect than ginger, probably due to the high-temperature resistance of the active ingredient in rosemary [14]. In addition, the formation mechanism of PAHs is related to free radicals [38]. The antioxidant substances contained in rosemary, such as phenols, may inhibit the formation of PAHs in fried pork balls by eliminating free radicals. However, it has also been well documented that antioxidants can promote oxidation at certain concentrations and under certain conditions [27]. Therefore, the increase in BaA, Chr, and BaP content in fried pork balls at 1.25% rosemary addition may be due to the pro-oxidant effect of the components in rosemary.

### 3.6. Effect of Spices on the Formation of TFAs in Deep-Fried Pork Balls

Table 5 shows the levels of C20:1 11t and C18:2 9t 12t and TFAs in fried pork balls. Three TFAs, C16:1 9t, C18:1 9t, and C18:1 11t, were not detected in any sample and therefore are not listed in the table. The spices showed a significant inhibitory effect on C20:1 11t in TFAs. The addition of ginger inhibited C20:1 11t from 16% to 40%, and the addition of rosemary inhibited C20:1 11t from 27% to 39%. The inhibition increased with increasing addition levels, but the difference in content was not statistically significant (*p* > 0.05). Pearson correlation analysis revealed that the C20:1 11t content of fried pork balls showed a positive correlation with TBARS values (r = 0.726, *p* ≤ 0.01) and a negative correlation with total phenol content (r = −0.928, *p* ≤ 0.01). Numerous studies have shown that cis-fatty acids can be converted to TFAs by isomerization during heating of fats and oils and that thermally induced isomerization is associated with lipid oxidation processes; in addition, lipid radicals are considered as intermediate products of isomerization reactions [39]. The antioxidant substances such as phenolic substances contained in ginger and rosemary may inhibit lipid peroxidation and unsaturated lipid isomerization reactions by scavenging free radicals generated during lipid oxidation. On the other hand, the inhibition of total TFAs by the addition of rosemary was stronger than that of ginger powder, and this result may be related to the strong thermal stability of the antioxidant active ingredients contained in rosemary, such as syringic acid [14].

## 4. Conclusions

This study investigated the effects of ginger and rosemary on the formation of HCAs, PAH4, and TFAs in deep-fried pork balls. The results showed that the addition of appropriate amounts of ginger and rosemary had different degrees of inhibitory effects on the formation of HCAs, PAH4, and TFAs in deep-fried pork balls. The addition of 0.25% ginger or rosemary decreased the total HCAs in the fried pork balls by 63% and 59%, respectively, compared with the control group, and the results showed that the lower spice addition concentration showed a higher inhibitory effect. For PAH4, the addition of ginger had no significant effect on the PAH4 content in fried pork balls, while the addition of 0.25% and 0.75% rosemary resulted in a significant reduction in PAH4 content in fried pork balls compared to the control group, which was attributed to the stronger antioxidant capacity of rosemary than ginger. The spices also showed significant inhibition of C20:1 11t in TFAs, with 40% and 39% inhibition of C20:1 11t for 1.25% ginger and rosemary, respectively. The combined results of the studies found that the best overall inhibition was achieved when 0.75% rosemary was added to the fried meatballs. The above results indicate that the addition of ginger and rosemary at appropriate concentrations to processed fried pork products can simultaneously reduce HCAs, PAH4, and TFAs, and the effect of rosemary was stronger than that of ginger in terms of the total inhibitory effect.

## Figures and Tables

**Figure 1 foods-11-03767-f001:**
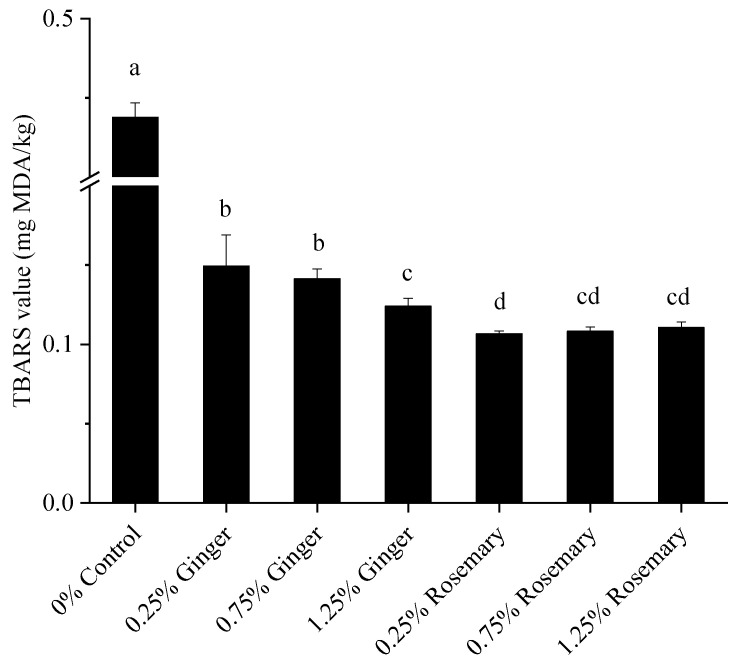
Effect of spice addition on the TBARS of deep-fried pork balls. Different small letters indicate significant difference at *p* < 0.05.

**Table 1 foods-11-03767-t001:** Antioxidant capacity and total phenolic content of spices.

Spice Name	Antioxidant Activity IC_50_ on DPPH Radicals (mg/mL)	Antioxidant Activity IC_50_ on ABTS Radicals (mg/mL)	Total Antioxidant Capacity (μmol/mL)	Total Phenolic (mg Gallic Acid Equivalents/g)
Ginger	0.26	4.60	0.58 ± 0.01	9.90 ± 0.27 ^a^
Rosemary	0.03	0.93	3.74 ± 0.04	61.34 ± 1.23 ^b^

Different small letters indicate significant difference at *p* < 0.05.

**Table 2 foods-11-03767-t002:** Cooking loss, moisture content, pH, and fat content of deep-fried pork balls.

	Concentrations (%)	Cooking Loss (%)	Moisture (%)	pH	Fat (%)
Control	0	33.00 ± 1.00 ^a^	55.28 ± 0.01 ^a^	6.64 ± 0.04 ^abc^	6.39 ± 0.14 ^a^
Ginger	0.25	30.17 ± 0.58 ^b^	58.14 ± 0.44 ^c^	6.72 ± 0.04 ^c^	6.27 ± 0.18 ^a^
	0.75	28.67 ± 0.29 ^c^	58.16 ± 0.13 ^c^	6.69 ± 0.06 ^bc^	6.74 ± 0.24 ^ab^
	1.25	26.33 ± 0.29 ^de^	58.28 ± 0.11 ^c^	6.57 ± 0.03 ^abc^	7.57 ± 0.50 ^b^
*p* (Concentrations)		**	ns	ns	**
Rosemary	0.25	30.83 ± 0.29 ^b^	56.73 ± 0.05 ^b^	6.47 ± 0.13 ^a^	7.57 ± 0.95 ^b^
	0.75	26.67 ± 0.76 ^d^	58.62 ± 0.14 ^d^	6.52 ± 0.11 ^ab^	7.41 ± 0.82 ^b^
	1.25	25.33 ± 1.04 ^e^	58.75 ± 0.08 ^d^	6.61 ± 0.06 ^abc^	7.41 ± 0.14 ^b^
*p* (Concentrations)		**	**	ns	ns
*p* (Spice)		*	ns	*	*

Results with different letters in the same column are significantly different at the level *p* < 0.05. ns means not significant (*p* > 0.05). * means *p* < 0.05; ** means *p* < 0.01.

**Table 3 foods-11-03767-t003:** Effect of spices on the formation of HCAs in deep-fried pork balls.

	Concentrations (%)	Norharman	Harman	IQ	IQx	Other HCAs	Total HCAs
Control	0	1.44 ± 0.21 ^a^	2.02 ± 0.18 ^a^	4.16 ± 0.16 ^a^	0.62 ± 0.04 ^a^	0.59 ± 0.11 ^a^	8.84 ± 0.43 ^a^
Ginger *p* (Concentrations)	0.25	1.65 ± 0.09 ^b^	0.69 ± 0.14 ^c^ (66%)	0.43 ± 0.12 ^b^ (90%)	Nq (100%)	0.53 ± 0.02 ^ab^ (10%)	3.29 ± 0.35 ^c^ (63%)
	0.75	2.65 ± 0.05 ^d^	0.74 ± 0.18 ^cd^ (63%)	0.66 ± 0.10 ^b^ (84%)	Nq (100%)	0.52 ± 0.04 ^ab^ (13%)	4.57 ± 0.36 ^bc^ (48%)
	1.25	4.04 ± 0.09 ^f^	1.16 ± 0.42 ^bd^ (43%)	1.77 ± 1.45 ^b^ (58%)	0.32 ± 0.16 ^b^ (48%)	0.54 ± 0.01 ^ab^ (9%)	7.83 ± 2.08 ^a^ (11%)
		**	ns	ns	**	ns	**
	0.25	2.01 ± 0.07 ^c^	0.39 ± 0.07 ^d^ (81%)	0.69 ± 0.91 ^b^ (84%)	Nq (100%)	0.50 ± 0.02 ^ab^ (15%)	3.59 ± 0.88 ^c^ (59%)
Rosemary	0.75	3.72 ± 0.09 ^e^	0.99 ± 0.13 ^bc^ (51%)	0.85 ± 0.46 ^b^ (80%)	Nq (100%)	0.46 ± 0.09 ^b^ (22%)	6.03 ± 0.54 ^b^ (32%)
	1.25	5.55 ± 0.10 ^g^	0.92 ± 0.11 ^bc^ (54%)	1.50 ± 0.63 ^b^ (64%)	0.26 ± 0.08 ^bc^ (57%)	0.53 ± 0.03 ^ab^ (11%)	8.41 ± 0.19 ^a^ (5%)
*p* (Concentrations)		**	**	ns	**	ns	**
*p* (Spice)		**	ns	ns	ns	ns	ns

The values are the means ± SD (ng/g fired pork balls sample). Results with different letters in the same column are significantly different at the level *p* < 0.05. Nq means not quantified; (%) means inhibition rate; ns means not significant (*p* > 0.05); ** means *p* < 0.01.

**Table 4 foods-11-03767-t004:** Effect of spices on the formation of PAH4 in deep-fried pork balls.

	Concentrations (%)	BaA	Chr	BaP	Total PAH4
Control	0	0.85 ± 0.20 ^b^	3.90 ± 0.23 ^ab^	4.59 ± 0.18 ^bc^	9.34 ± 0.27 ^b^
Ginger *p* (Concentrations)	0.25	0.62 ± 0.07 ^a^ (27%)	3.55 ± 0.32 ^b^ (9%)	5.33 ± 0.48 ^ab^	9.50 ± 0.77 ^b^
	0.75	0.92 ± 0.08 ^bc^	3.98 ± 0.07 ^ab^	4.18 ± 0.41 ^c^ (9%)	9.08 ± 0.52 ^b^ (3%)
	1.25	0.81 ± 0.07 ^ab^ (5%)	3.84 ± 0.39 ^ab^ (2%)	4.62 ± 0.67 ^bc^	9.26 ± 1.12 ^b^ (1%)
		**	ns	ns	ns
	0.25	0.87 ± 0.18 ^b^	2.52 ± 0.36 ^c^ (35%)	3.18 ± 0.70 ^d^ (31%)	6.57 ± 0.95 ^a^ (30%)
Rosemary	0.75	1.13 ± 0.13 ^c^	2.30 ± 0.14 ^c^ (41%)	2.64 ± 0.16 ^d^ (42%)	6.07 ± 0.36 ^a^ (35%)
	1.25	1.13 ± 0.04 ^c^	4.28 ± 0.27 ^a^	5.52 ± 0.30 ^a^	10.93 ± 0.24 ^d^
*p* (Concentrations)		ns	**	**	**
*p* (Spice)		**	**	**	**

The values are the means ± SD (ng/g fired pork balls sample). Results with different letters in the same column are significantly different at the level *p* < 0.05. (%) means inhibition rate; ns means not significant (*p* > 0.05); ** means *p* < 0.01.

**Table 5 foods-11-03767-t005:** Effect of spices on the formation of TFAs in deep-fried pork balls.

	Concentrations (%)	C20:1 11t	C18:2 9t 12t	Total TFAs
Control	0	3.30 ± 0.26 ^a^	3.89 ± 0.42 ^a^	7.20 ± 0.68 ^bc^
Ginger *p* (Concentrations)	0.25	2.76 ± 0.12 ^ab^ (16%)	5.51 ± 1.04 ^c^	8.61 ± 0.50 ^d^
	0.75	2.31 ± 0.69 ^bc^ (30%)	5.13 ± 0.69 ^bc^	7.44 ± 1.31 ^cd^
	1.25	1.99 ± 0.23 ^c^ (40%)	3.95 ± 0.45 ^ab^	5.94 ± 0.67 ^ab^ (17%)
		ns	*	*
	0.25	2.43 ± 0.24 ^bc^ (27%)	4.14 ± 0.29 ^ab^	6.56 ± 0.53 ^abc^ (9%)
Rosemary	0.75	2.30 ± 0.12 ^bc^ (30%)	3.86 ± 0.08 ^a^ (1%)	6.16 ± 0.17 ^abc^ (14%)
	1.25	2.01 ± 0.08 ^c^ (39%)	4.25 ± 0.85 ^ab^	5.84 ± 0.42 ^a^ (19%)
*p* (Concentrations)		ns	ns	ns
*p* (Spice)		ns	**	**

The values are the means ± SD (mg/100g fired pork balls sample). Results with different letters in the same column are significantly different at the level *p* < 0.05. (%) means inhibition rate; ns means not significant (*p* > 0.05); * means *p* < 0.05; ** means *p* < 0.01.

## Data Availability

The data used to support the findings of this study can be made available by the corresponding author upon request.

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
