# Peer review of "Inhibiting Effects of Ginger and Rosemary on the Formation of Heterocyclic Amines, Polycyclic Aromatic Hydrocarbons, and Trans Fatty Acids in Fried Pork Balls"

_foods, 2022, doi:10.3390/foods11233767_

Round 1

Reviewer 1 Report

1. IARC Monographs on the identification of carcinogenic hazards to humans include much more PAHs (https://monographs.iarc.who.int/list-of-classifications) than those mentioned in this paper. IARC should be included as reference. 

2. Benz[a]anthracene, chrysene, benzo[b]fluoranthene, and benzo[a]pyrene are part of PAH4, but according to Commission Regulation (EU) No 835/2011. 

2. Could you add validation of methods (LOD, LOQ, recovery...)?

3. Janoszka et al examined different PAHs (Fln, BaA, BaP, BkFln, BghiP) than you, so it is not suitable for comparison. 

4. The addition of rosemary did not caused a significant increase in BaA content in all concentrations (control group 0.85, with 0.25 rosemary 0.87- there is no significant increase). So you have to change comparison with Lu et al. (2018). 

5. Inhibitory rate for PAH4 with 0,75% is 35% , not 45% not 45% as stated in the text (367th row).

6. It would be great to state in the conclusion - which  concentration of  rosemary gives the best results, taking into account all the tests.

Reviewer 2 Report

In the present manuscript, the authors deal with a current problem related to the safety of meat products. Red meat itself was listed at IARC in Class 2A and the formation of high-risk substances is also associated with the technological treatment of meat. People are starting to try to reduce this risk without having to give up their favorite traditional foods, and one way is to use natural antioxidants from herbs. The chosen topic is therefore up to date and the results will certainly find practical application. However, the other methods of reduction of harmful compounds shoud be mentioned in the literature overview.

The article describe analyses of a whole range of compounds that have an adverse effect on the human organism (PAHs, HCAs..). The truth is, that the most of the harmful compounds come from the frying medium – and this should be also discussed in the article. There is lot of current literature published in last two years that should be cited in the literature overview. Random some articles that should not be ommited in the Introduction and Discussion parts:

Review of the authors J. Shen et al. Published in Critical Reviews in Food Sciences and Nutrition which summarizes strategies for increasing antioxidant aktivity. (https://doi.org/10.1080/10408398.2021.2019672)

Recently published article of the autors Zhu et al. In Food Control dealing with PAHs detection and elimination (https://doi.org/10.1016/j.foodcont.2022.109194)

Dutta et al. published different protection strategy –coating fried material with edible films (in Food Control https://doi.org/10.1016/j.foodcont.2022.109194)

Suggested minor modification and revisions:

A clearer explanation of the preparation and frying procedure on lines 109-110 woud be appropriate. I understand that the fryer was preheated and meat balls were fried using soybean oil at 180 °C for 3 min. (It is better to use Past progressive – the procedure was done in the past, applicable in other parts of the methods.)

Line 204 –Weighted, typos; and overal construction of this sentence should be modified.

More attention should be paid to correct citations. E.g. cit. No 4 (line 444) should be:

Onopiuk A, Kołodziejczak K, Szpicer A, Wojtasik-Kalinowska I, Wierzbicka A, Półtorak A. Analysis of factors that influence the PAH profile and amount in meat products subjected to thermal processing. Trends in Food Science & Technology. 2021, 115, 366-79.

Question:

It would be interesting to see the effect of ginger and rosemary together, have you tried the combination of the herbs?
